# Differences in Data Trustworthiness and Risk Perception between Bar Graphs and Pictograms

**DOI:** 10.3390/ijerph19084690

**Published:** 2022-04-13

**Authors:** Munehito Machida, Michio Murakami, Aya Goto

**Affiliations:** 1Center for Integrated Science and Humanities, Fukushima Medical University, Fukushima 960-1295, Japan; agoto@fmu.ac.jp; 2Department of Health Risk Communication, Fukushima Medical University, Fukushima 960-1295, Japan; michio@cider.osaka-u.ac.jp; 3Center for Infectious Disease Education and Research (CiDER), Osaka University, Osaka 565-0871, Japan

**Keywords:** health literacy, pictogram, bar graph, risk perception, reliability

## Abstract

We investigated whether differences in presentation style affect risk perception, understanding, preference, and trust toward data. One hundred and sixty Fukushima Medical University students were shown the lifetime probability of breast cancer incidence for a 50-year-old woman, presented in both a pictogram and a horizontal bar graph format. Participants rated each of the following on a five-point scale by looking at each figure: risk perception, perceived truth of data, and comparative risk perception. The perceived truth of data was high for pictograms, especially among men and among those defined as having lower health literacy. Women correctly perceived the risk of breast cancer as higher than that of dying in a car accident when the data were presented on a pictogram. There was no difference in risk perception, perceived truth of data, or comparative risk perception arising from being shown the bar graphs and the pictograms in a particular order. There was a 50/50 split on which type of graph was perceived as easier to understand, but the preference was for the pictogram format. It is important to devise a visual method of health communication that considers the purpose of the information and characteristics of the target audience.

## 1. Introduction

In 2011, Japan was hit by the Great East Japan Earthquake and Tsunami, which caused extensive damage mainly along the Pacific coast of the Tohoku region. In Fukushima Prefecture, the earthquake and tsunami triggered an accident at the Fukushima Daiichi Nuclear Power Station, and immediately after the release of radioactive materials, Fukushima residents were plagued with anxiety about the health effects of radiation. In order to respond to the residents’ concerns and needs for health information, the local governments within Fukushima Prefecture, together with Fukushima Medical University, have provided health education on radiation to the residents and have made continuous efforts to spread the correct knowledge and alleviate their concerns [1]. The medical university continues to provide training in health literacy and risk communication skills to public health nurses at public health centers who work on health promotion and health education for residents [2]. Health literacy is defined by the World Health Organization as the knowledge and skills to obtain, understand, and use information to maintain and improve health [3]. More recently, the skills of health professionals to convey health information have also become an important part of health literacy [4].

At Fukushima Medical University, “Social Communication” was introduced into the lecture series on social medicine at the School of Medicine, based on the abovementioned experiences on radiation risk communication and health literacy workshops in the field. Students additionally have the opportunity to learn health literacy and risk communication as part of a science literacy laboratory in the School of Medicine and as part of epidemiology and biostatistics classes and a health information laboratory in the School of Nursing [5]. After they graduate from university, one of their core roles as health professionals will be to explain to patients and residents the benefits and risks of the medical services they provide. However, such an opportunity for medical and nursing students to learn about health literacy and risk communication is scarce in medical education in Japan [6]. In the course of the region’s long-term recovery, Fukushima Medical University has established distinctive curricula that reflect the need for health professionals to communicate with local communities about low-dose radiation exposure [5].

Recent studies reported that patients showed a better understanding of numerical data with appropriate use of visual aids such as pie charts, icon arrays, bar charts, and pictograms [7,8]. Yet, we should be aware that effective ways of presenting charts and graphs differ depending on the characteristics of the target audience, and that the perception of risk varies depending on the type and method of visual presentation [9,10]. In the abovementioned science literacy laboratory at Fukushima Medical University, medical students conducted a survey among their peers to analyze the changes in risk perception depending on the method of data presentation. Using the data collected in the exercise, we investigated whether the format of data presentation (bar graph or pictogram) affected risk perception and trust in the data, if the magnitude of influence differed depending on gender, health literacy level, and order of presentation, and further, which presentational format was preferable among students.

## 2. Materials and Methods

During the 2018–2019 academic year, 16 research topics were presented to approximately 140 first-year medical students in each year in Fukushima Medical University’s science laboratory class. The students were asked to list their first to third choices of the class topics and were then allocated to 16 groups accordingly. This study used the data collected by 16 students in 2 years who were allocated to the health literacy group. These students each distributed self-administered questionnaires to 10 friends who belonged to the School of Medicine and School of Nursing in the same university and then collected the completed responses. Thus, the selection criterion for the target population of this study is that they were friends of students in the health literacy laboratory class and belonged to the School of Medicine or School of Nursing at Fukushima Medical University.

For the questionnaire, data from the study by Schapira and colleagues [11] that showed the lifetime probability of breast cancer incidence (9%) for a 50-year-old woman were presented in two different ways: as a pictogram, and as a 100% stacked horizontal bar graph, followed by the same questions on risk perception as in Schapira’s study [11]. These figures represent the risk of breast cancer incidence as a typical single event in a population. Participants were asked to look at the data and to score the following factors on a scale of 1–5 points for both presentational formats: (1) risk perception, as the risk of you or someone close to you getting breast cancer, scored from very low (1) to very high (5); (2) perceived truth of data, as the extent to which you feel you can trust the data in the format they are presented, scored from very low (1) to very high (5); and (3) comparative risk perception, as the probability of getting breast cancer compared with the probability of dying in a car accident, with the options 1/100 (1), 1/10 (2), the same (3), 10 times higher (4), and 100 times higher (5). The method of asking the comparative risk of events using death in a car accident as a reference point has been used in a previous study [12]. Because these questions ask how the risk is perceived, they do not require specific knowledge to answer. In terms of the comparative risk perception question in our study, the most appropriate answer is between 4 and 5, i.e., the probability of getting breast cancer is between 10 and 100 times higher than that of dying in a car accident. In addition, half of the participants saw the pictograms first and the bar graphs later, while the other half saw them in the reverse order, enabling us to examine the potential for differences in risk perception based on the order in which the two data presentation formats were viewed.

The approach described in Ishikawa’s Communicative and Critical Health Literacy (CCHL) [13] was used to determine the health literacy level of each participant. The CCHL measurement method has been commonly used in previous studies [14,15]. Respondents were asked whether they would be able to (1) collect health-related information from various sources, (2) extract the information they wanted, (3) understand and communicate the obtained information, (4) consider the credibility of the information, and (5) make decisions based on the information, specifically in the context of health-related issues. Participants were asked to self-rate their abilities with respect to each of these items on a 5-point scale, from very low (1) to very high (5). Additionally, in the questionnaires conducted in 2019, 80 participants were asked further questions about their preference for the pictograms or the bar graphs and their respective ease of understanding.

For statistical analysis, JMP Pro 11.2.0 was used (SAS Institute, Cary, NC, USA). In the analysis, *p* < 0.05 was considered statistically significant. We first examined the distribution of the risk perception and perceived truth of data by presentation format (pictogram and bar graph). Then, within-group differences in the distribution stratified by gender and health literacy level were examined using the Wilcoxon signed-rank test. Lastly, differences arising from the order in which the two data presentation formats were shown to participants were examined using the Mann–Whitney U test.

Because this study is a secondary analysis of existing data collected anonymously, Fukushima Medical University exempted ethical review in accordance with the ethical guidelines for medical research on human subjects by the Ministry of Education, Culture, Sports, Science and Technology and the Ministry of Health, Labour and Welfare.

## 3. Results

There were 160 participants (98 men and 62 women), mostly (n = 146) affiliated with the School of Medicine (years 1–6) (Table 1). We confirmed that there were no invalid responses or missing data, except for the absence of academic year and school affiliation information for one student.

Regarding the difference between pictograms and bar graphs, the perceived truth of data tended to be higher for pictograms than for bar graphs (*p* = 0.06). However, there was no difference between the graphs in terms of risk perception and comparative risk perception (Table 2).

Stratifying by gender (Table 3) revealed that, for men, the perceived truth of data score for pictograms was significantly higher than for bar graphs (*p* = 0.03), i.e., men were significantly more likely to place trust in data presented in a pictogram than in the same data presented in a bar graph. For women, the comparative risk perception score for the pictogram tended to be higher than for the bar graph (*p* = 0.07). This means that women tended to be more likely to correctly interpret the comparative risk (i.e., the probability of getting breast cancer relative to that of dying in a car accident) when viewing the data on a pictogram compared with viewing the same data on a bar graph. To stratify participants by health literacy level (Table 4), we dichotomized them using the mean of their self-assessed CCHL scores across all respondents, which was 3.3. We therefore defined a “higher health literacy” group, containing all respondents with an average self-rated score of 3.3 or higher across the five health literacy questions, and a “lower health literacy” group, containing all respondents with an average self-rated score below 3.3 across the health literacy questions. In the lower health literacy group, the truth of the data was perceived as significantly higher for the pictogram (*p* = 0.03) compared with the bar graph. There were no significant differences in risk perception or perceived truth of data between the groups that were shown the bar graphs and the pictograms in a different order (Table 5). 

In reviewing the data obtained by asking supplementary questions to 80 participants in 2019, we observed a 50/50 split on which data presentation format was easier to under-stand, but 67.5% (54/80) of respondents stated a preference for the pictogram format. There was no significant difference in the preference for pictograms or bar graphs (*p* = 0.27), or in the perceived ease of understanding (*p* = 0.37), reported by men and women.

## 4. Discussion

In this study, we analyzed the differences in risk perception, perceived truth of data, and comparative risk perception when pictograms and bar graphs containing the same information were presented. Figure 1 shows the summary of our study findings. First, pictograms tended to be perceived as more reliable than bar graphs, especially by men and those with lower health literacy. Recent studies conducted among parents [16] and a broad age range of the general public [17] reported that the pictogram was perceived as more trustworthy than graphs. In Schapira’s study [11], no difference in perceived truth of data was found between pictograms and bar graphs among women with a different age distribution (40 to 85 years) to that in our study. With respect to numeracy levels, pictograms were reported to be a desirable option across all levels for achieving adequate knowledge from the presented information [17]. Second, women were able to more accurately perceive the risk of getting breast cancer compared with that of dying in a car accident when shown the data in a pictogram. Interestingly, the survey of parents reported that those who received pictographic information perceived the risks to be lower and the benefits to be higher compared with those who received the same information in different formats [16]. Third, our respondents preferred pictograms over bar charts. Schapira’s study and another previous study reported the same finding [11,18].

Visual aids have been proposed as a way to support the understanding of quantitative risk. It should be noted that pictorial presentation may influence the interpretation of risk information even among university students of health professions. These students, who see health information daily, preferred the pictographic presentation. Of note, their mean CCHL score of 3.3 was comparable to that of the Japanese general public (3.7) [13]. In recent years, various studies have been conducted to examine the impact of the pictorial format on the understanding of health information [19]. Bar graphs and pictograms were the top two visual aids that were studied and shown to improve patients’ understanding and satisfaction [20,21]. Pictograms, in particular, which are often the preferred tool for use in patient education, have been applied in various types of interventions including discharge instruction education, risk communication, and medication education. The authors of a review on pictogram usage warned that pictograms should be designed carefully in accordance with the purpose and content of the information [19]. Further, information should be designed for and tailored to the targeted audience. For example, people with a higher education level may benefit less from the graphic presentation of information [22]. Ancker and colleagues stated that communicators should not assume that all graphics are more intuitive than text; many of the studies found that patients’ interpretations of the graphics were dependent upon expertise or instruction [23].

Furthermore, cultural differences should also be considered with regard to differences in visual understanding. On a positive note, a previous study showed that when only numerical data were presented, there was a large difference in comprehension among subjects of different nationalities, but when a pictogram was also presented at the same time, the difference became smaller [24]. However, another study reported that reactions toward visualizations can also be culture-specific; for example, in America, red is often associated with loss, whereas in China, green is often associated with loss, and therefore the use of these colors in visual aids intended for an audience with a particular cultural background can elicit a higher degree of risk aversion in their reaction to the data presented [25]. Related to culture-specific differences, we also observed gender differences in risk perception and perceived truth of data in our study. Despite the increasing research attention on graph literacy in recent years, the literature has thus far been inconclusive on such gender differences [26,27]. It is easy to imagine that cultural and gender backgrounds are interwoven in risk perception [28]. Future research should consider interactions among the information content, data presentation, and characteristics of the audience to explore improved ways to communicate health information.

Overall, our study and the results of the previous studies mentioned above highlight that commonly preferred pictograms increase perceived trust in communicating risk information to men and those with low health literacy, and further, they help women to accurately capture the magnitude of risk. As for practical implications, these results demonstrate the effectiveness of the selective use of pictograms according to the characteristics of the target population. It should also be noted that the choice depends on the purpose of the communication. The purpose of risk communication is not necessarily only to increase perceived trust or provide an accurate understanding of the magnitude of the risk. In environmental and public health issues, such as the Fukushima nuclear disaster, risk communication has been reported to have a variety of purposes, including decision-making support, stress reduction, and value sharing, as well as trust and understanding [29]. The risk communication method should be based on a deep consideration of its purpose and the fact that the presentation of different types of graphs can have different effects depending on the characteristics of the target group. This requires bilateral communication between health professionals and the general public. In health professional education, students should be aware of the importance of health literacy and improve skills to better convey health information.

There were three major methodological limitations in the present study. First, we only compared pictograms and bar graphs, in reference to Schapira’s study [11], despite other various types of visual materials, such as pie charts and infographics, also being commonly used. It is desirable to compare various graphical formats in further research. Second, the health literacy level of participants was assessed by a single self-assessment measure. Objective and educational measurements such as the Newest Vital Sign (a health literacy scale that asks people to read a nutritional label on an ice cream cup) could be added in future research [30,31]. Third, the sample size was small, with 160 participants in total. In addition, they were recruited from students’ peers at the School of Medicine and School of Nursing, which presents a potential selection issue. The students at a health professional university may have more interest in health literacy than other undergraduate students. Both of these factors limit the generalizability of our obtained data. Yet, evidence is scarce in Japan with regard to the influences of data presentation on risk perception, and our study indicates a need for further academic efforts on the topic in the country.

In the aftermath of a health crisis such as the radiation disaster in Fukushima, risk communication skills are required for health professionals to provide health care information to residents in an easy-to-understand way. It is essential that these skills are also developed as part of the educational process [32].

## 5. Conclusions

Our preliminary data collected among students of health professions presented varying effects of visual presentation of health information on the audience’s risk perception, which differed depending on their gender and health literacy level. Taking advantage of the existing health communication education opportunities offered to students at Fukushima Medical University, we will continue to explore ways of devising effective visual methods of data presentation that consider the purpose of the information and characteristics of the target audience, including their cultural differences.

## Figures and Tables

**Figure 1 ijerph-19-04690-f001:**
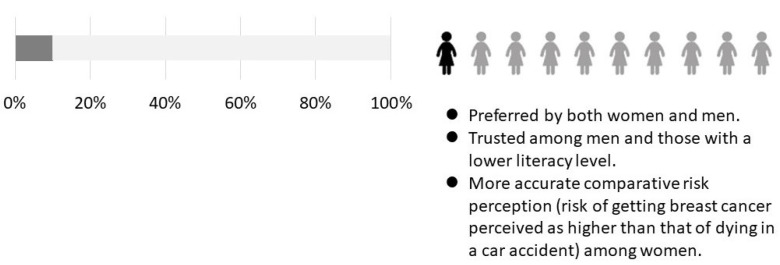
Summary of main findings.

**Table 1 ijerph-19-04690-t001:** Characteristics of respondents.

	Characteristics		n
Year	School of Medicine	1	125
		2	1
		3	5
		4	10
		5	0
		6	5
	School of Nursing	1	10
		2	2
		3	1
		4	0
	Unknown		1
Gender	Men		98
	Women		62

**Table 2 ijerph-19-04690-t002:** Distribution of risk perception and data reliability by data presentation format.

Survey Items				N (%)			Mean Score			
			Pictogram			Bar Graph	Pictogram		Bar Graph	*p-Value*
The risk of you or someone close to you getting breast cancer										
(very low)	1	2	0.41%		8	1.66%	3.0		3.0	0.68
	2	58	23.82%		53	21.95%				
	3	43	26.49%		41	25.47%				
	4	45	36.96%		44	36.44%				
(very high)	5	12	12.32%		14	14.49%				
The perceived truth of the data										
(very low)	1	4	0.85%		10	2.21%	2.9		2.8	0.06
	2	45	19.07%		41	18.14%				
	3	74	47.03%		80	53.10%				
	4	29	24.58%		25	22.12%				
(very high)	5	8	8.47%		4	4.42%				
The probability of getting breast cancer compared with that of dying in a car accident										
(1/100)	1	7	1.33%		6	1.15%	3.3		3.3	0.66
(1/10)	2	40	15.15%		44	16.83%				
(same)	3	40	22.73%		39	22.37%				
(10 times)	4	44	33.33%		43	32.89%				
(100 times)	5	29	27.46%		28	26.77%				

Wilcoxon signed-rank test was used.

**Table 3 ijerph-19-04690-t003:** Differences in risk perception and data reliability depending on data presentation style: stratified by gender.

Survey Items	Mean Score	
	Men n = 98	Women n = 62
The risk of you or someone close to you getting breast cancer		
Pictogram	3.1	2.9
Bar graph	3.1	2.9
*p-value*	*0.51*	*0.89*
The perceived truth of the data		
Pictogram	2.9	3.0
Bar graph	2.8	2.9
*p-value*	*0.03*	*0.52*
The probability of getting breast cancer compared with that of dying in a car accident		
Pictogram	3.3	3.3
Bar graph	3.4	3.1
*p-value*	*0.29*	*0.07*

Wilcoxon signed-rank test was used to examine differences between scores for the pictogram and bar graph.

**Table 4 ijerph-19-04690-t004:** Differences in risk perception and data reliability depending on data presentation style: stratified by health literacy level.

Survey Items		Mean Score	
		High n = 29	Low n = 131
The risk of you or someone close to you getting breast cancer			
Pictogram		3.1	3.0
Bar graph		3.2	3.0
*p-value*		*0.39*	*0.72*
The perceived truth of the data			
Pictogram		2.9	3.0
Bar graph		2.9	2.8
*p-value*		*0.71*	*0.03*
The probability of getting breast cancer compared with that of dying in a car accident			
Pictogram		3.7	3.2
Bar graph		3.7	3.2
*p-value*		*0.68*	*0.39*

Wilcoxon signed-rank test was used to examine differences between scores for the pictogram and bar graph.

**Table 5 ijerph-19-04690-t005:** Differences in risk perception and data reliability depending on the order in which data presentation styles were seen.

Survey Items	Mean Score			
	Pictogram →Bar Graph (n = 80)	Bar Graph →Pictogram (n = 80)	*p*-*Value*	
The risk of you or someone close to you getting breast cancer				
Pictogram	3.1	3.0	0.92	
Bar graph	3.1	2.9	0.16	
The perceived truth of the data				
Pictogram	3.0	2.9	0.31	
Bar graph	2.9	2.8	0.35	
The probability of getting breast cancer compared with that of dying in a car accident				
Pictogram	3.2	3.4	0.17	
Bar graph	3.2	3.4	0.19	

Mann–Whitney U test was used.

## Data Availability

The data presented in this study are available on request from the corresponding author.

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
