# Peer review of "Differences in Data Trustworthiness and Risk Perception between Bar Graphs and Pictograms"

_ijerph, 2022, doi:10.3390/ijerph19084690_

Round 1

Reviewer 1 Report

The great value of the work is emphasizing the role of health literacy.

The article is an important part of the discussion on the effectiveness of prophylaxis and health education.
It is important to look for new and effective ways of education, especially in the context of diseases where early detection allows for cure.

However, the results of the study should be interpreted taking into account the cultural determinants of health.
Therefore, it is important that similar tests should be also carried out in other countries and this should be completed with the conclusions.

In conclusion, I recommend taking into account the probable differences in outcomes in the context of cultural differences

Reviewer 2 Report

This work designed a questionary to show 160 students the lifetime probability of breast cancer incidence for a 50-year-old woman, using a pictogram and bar chart, hence, to discover the difference between those two presentation graph styles based on the scale measures for each graph. It is an interesting study but comes with many flaws. I cannot see the contributions now.

Firstly, I understand it is work regarding public health, but it also addresses the Differences in data reliability and risk perception between two types of graphs. However, this study uses outdated references when graph drawing has been developing at a high rate, and I suggest citing more latest research.

Secondly, there are tons of graph drawing studies, so what motivated the authors to choose bar and pictograms, and what are the pros and cons of those two visual presentations compared to others?

Also, regarding the questionary, "the lifetime probability of breast cancer incidence for a 50-year-old woman", here, does it mean only one patient involved, and all graphs generated were all from that person? What are the selection criteria for choosing students involved? How to ensure the questionaries were adequately answered? Are there any invalid questionaries? Is relevant knowledge essential to answer the question?

Besides, "especially by men and those with lower health literacy", what is the insights of this statement? Gender seems to be a factor that needs to be considered, and also health literacy, but without further details, I don't know what can be concluded.

I'd also like to see those graphs instead of offering several tables.

As the authors mentioned, "It is important to devise an appropriate visual method of health communication that reflects the characteristics of the target audience. Training future health professionals to be attentive to clients' information needs and characteristics when communicating with them would lead to building resilience in risk communication for current and future health crises", however, this study has severe limitations at the stage, such as the data sample, questionary details, experiment results etc. I am not convinced by the present format.

Reviewer 3 Report

This is an interesting and important study concerning risk communication, which is the main part of shared decision-making. The methodology and data analysis were very well described.

However, there are some issues:

  1. The sample consisted of medical/nursing students. First, they are educated people, and secondly, they are medical profession, therefore they might be different concerning the understanding of more abstract information (bar graphs, for example), and certainly, they have a better health literacy than usual people, and also than students in other domains. This is the main limitation of the generalizability of the study. Maybe a comparison could be done between nurses and medical students, and between students in their first years of study compared with those in later years of study, supposing that health literacy increases by the year of study.
  2. Health literacy was determined by a questionnaire which had a validation study, but for which the construct validity is not convincing (the results of the questionnaire were not compared with any objective measure of health literacy, but with health-related behaviors, coping management, and the experienced somatic symptoms, which in fact could only partially reflect health literacy).
  3. The participants rated the graph data by “risk perception, data reliability, and comparative risk perception”. In fact, it was the data reliability perception, as the participants rated the extent to which they trusted the data; data were reliable, I suppose, in both kinds of graphs.

I would have liked to see the presentations (both as bars and as pictograms, and both for the cancer risk and car accident).

Round 2

Reviewer 2 Report

I appreciate the authors' responses, but I don't think they have addressed most issues. There might be misunderstandings. I know it's an initial step of this study. However, I wanted to see clear content to explain, not just one statement, especially for those questions in my previous comments below.

  • Also, regarding the questionary, "the lifetime probability of breast cancer incidence for a 50-year-old woman", here, does it mean only one patient involved, and all graphs generated were all from that person? What are the selection criteria for choosing students involved? How to ensure the questionaries were adequately answered? Are there any invalid questionaries? Is relevant knowledge essential to answer the question?
  • Besides, "especially by men and those with lower health literacy", what is the insights of this statement? Gender seems to be a factor that needs to be considered, and also health literacy, but without further details, I don't know what can be concluded. I'd also like to see those graphs instead of offering several tables.

The authors did not answer those at all, just repeated some statements, and offered a figure which cannot get any info from. If this work was related to some visualization tech, the authors have to at least generate a better graph to prove their points. The figure in the present format is far from that aspect.

Reviewer 3 Report

the authors added all the weaknesses in the Discussion section/Limitations.
